# Spiritual Care: A Description of Family Members’ Preferences of Spiritual Care Nursing Practices in Intensive Care Units in a Private Hospital in Kwa-Zulu Natal, South Africa

**DOI:** 10.3390/healthcare10040595

**Published:** 2022-03-22

**Authors:** Mercy Zambezi, Waheedha Emmamally, Nomaxabiso Mooi

**Affiliations:** 1School of Nursing and Public Health, College of Health Sciences, University of KwaZulu-Natal, Durban 4000, South Africa; 214581940@stu.ukzn.ac.za; 2Department of Nursing, Faculty of Health Sciences, Walter Sisulu University, Mthatha 5099, South Africa; nmooi@wsu.ac.za

**Keywords:** family members, spiritual care, nursing practice, intensive care unit

## Abstract

Background: Spiritual care is a part of the holistic care that enables family members in intensive care units to find meaning in their life events and simultaneously bolsters their resilience and coping tools. Objective: To determine family members’ preferences of spiritual care practices that they require from nurses working in intensive care units. Methods: A quantitative, descriptive, cross-sectional study was conducted in the intensive care units of a private hospital in the province of KwaZulu-Natal. Data, using the Nurse Spiritual Therapeutic Scale, were collected from a purposive sample of family members (*n* = 47). Data were analyzed using descriptive statistics. Results: The mean overall Nurse Spiritual Therapeutic Scale was 58.4 (20–80). The most preferred and least preferred spiritual care practices by family members were “to be helped to have quiet time and space”, (*M* = 3.32, SD = 0.59) and “to arrange for a chaplain to visit them” (*M* = 2.70, SD= 0.91), respectively. Conclusion: The mean overall NSTS score indicated that there was a strong preference among family members for nurses to provide them with spiritual care in the intensive care units. However, due to the diversity of family members’ preferences it remains important that family members guide intensive care nurses in their spiritual care.

## 1. Introduction

An admission of a close family member to an intensive care unit (ICU) raises complex spiritual needs and questions in family members [1,2]. Often a family’s religious background or non-religious spiritual beliefs are valuable coping tools for dealing with the grief and redefining hope [3]. Indeed, family members who experience critical illness often turn to spirituality or religion for support [4]. Spirituality refers to a relationship with a higher power, or a response to a desire for self- transcendence [5,6] and is expressed through beliefs, values, traditions and practices [7]. According to Turan and Yavuz Karamanoğlu [8] spirituality includes, but is not limited to, religion and encompasses a search for the purpose and meaning of life. The difference between religion and spirituality may be hard to distinguish, and may overlap for people, and for these reasons the two concepts should be defined by the individual [9]. 

Spirituality is likened to armor against any negative emotional and psychological consequences experienced in a critical situation [10]. The authors add that family members with close family members in ICUs may seek solace and strength in their spiritual beliefs. In the ICUs, family members may turn to intensive care nurses to provide answers to and direction for their spiritual concerns [11]. Positive healthcare outcomes of improved decision-making, increased patient and family satisfaction with care and increased family coping were linked to the provision of spiritual care to family members of critically ill patients [12,13]. Notably, a number of studies have found that intensive care nurses experience challenges in providing spiritual care to family members [14,15]. Central to these challenges are the uncertainties of nurses on the spiritual care required by family members, who are so diverse in their spiritual orientations [16]. It would seem that inviting family members to guide us on their preferences regarding spiritual care practices would enable intensive care nurses to align spiritual care in critical care contexts to the family members’ spiritual worldviews.

Healthcare organizations subscribe to a philosophy of holistic care, emphasizing both family-centered and patient-centered care, but rarely focuses specifically on spiritual care for patients and family members [17]. At present, spiritual care activities that are in place are based on the assumptions and perspectives of healthcare providers regarding what patients and family members require in terms of spiritual support [18]. Hence, there exists the need to determine family members’ preferences for the spiritual care practices that they require from the nurses working in ICUs. 

The objective of the current study was to determine family members’ preferences of the spiritual care practices that they require from the nurses working in ICUs.

## 2. Materials and Methods

Design and setting: A quantitative, descriptive cross-sectional study was conducted in three ICUs (medical, surgical, and cardiac) of a private hospital in the province of KwaZulu-Natal, South Africa (SA). The South African healthcare system comprises of state-funded and private healthcare sectors, with the latter providing healthcare services to people who are able to pay privately for the services or who have health insurance [19]. It is likely that patients and families who pay for the health care services rendered have greater expectations of comprehensive care that includes spiritual care than patients receiving care at state-funded health hospitals [20]. 

Study population, sampling and sample: This study targeted family members of patients admitted into the selected ICUs. The study received permissions from the ethical commitees and gate-keepers in February 2020, at the time when SA began to confront the COVID-19 pandemic. In anticipation of the hospital visiting hours being curtailed, the researchers, in consultation with a statistician, made a decision to survey all family members who fit the eligibility criteria. The eligible family members were purposively sampled using the Total Population Sampling (TPS). TPS involves the inclusion of the entire population meeting the criteria in the study being conducted and is commonly used where the number of cases being investigated is relatively small, which was the case in the current study [21]. The study had sampled 47 respondents when SA went into lockdown on the 26 March 2020 and data collection was curtailed.

Inclusion and exclusion criteria: The inclusion criteria for this study included that family members: (i) were identifed by themselves or the patient as being significant to the patient; (ii) were over the age of 18 and willing to participate in the study; and (iii) were able to complete the English- or isiZulu-translated questionnaire. Excluded family members were those who met the inclusion criteria but were unwilling to participate in the study. 

Measuring instruments: Data were collected from family members using a self-administered questionnaire, comprising of questions on the sociodemographic characteristics of the respondents (gender, age, relationship to patient, religion) and the Nurse Spiritual Therapeutic Scale (NSTS) [22]. The study focused on the 20 items of the NSTS which relate to clients’ preferences regarding spiritual care practices that they receive from nurses. Examples of the practice items include: In general, I would want my family member’s nurse to: “*Help me to have quiet time and space*”, and “*Help me laugh (e.g., share a joke)*”.

Each item (spiritual care practice) scored on a 4-point Likert scale were from 1 = “strongly disagree” to 4 = “strongly agree”. The possible range of response is 20–80. A score of 20 indicates strong disagreement about wanting a nurse to provide the spiritual care practices identified in the NSTS, whereas a high score of 80 indicates the desire for a nurse to provide spiritual care practices [23]. The NSTS was translated, with permission from the developers, to isiZulu which is the spoken-language of the majority of family members visiting the selected hospital. Face validity of the English version of the NSTS was confirmed by the supervisor who is a clinical critical care nurse specialist, whilst the isiZulu version of the NSTS was confirmed by an isiZulu-speaking clinical nurse working in the ICU setting. The NSTS has an established high internal reliability of 0.97 [22]. In the current study, the researchers used two pilot studies of five family members each to determine the reliability of the isiZulu and English versions of the questionnaire. Both the isiZulu and English versions of the questionnaire showed strong internal consistency of Cronbach’s alpha = 0, 92 and 0.94, respectively. The results of the pilot study were not included in the main study and no changes were made to the NSTS.

Study variables: Study variables included sociodemographic variables (gender, age, religion and relationship to patient) and preferred spiritual care practices.

Study procedures: After obtaining ethical approval and permission from the hospital authorities, data collection commenced from 1 March 2020 and stopped abruptly on 26 March 2020, when hospital visitations were stopped due to the COVID-19 pandemic. During data collection, COVID-19 precautions included the use of personal protective equipment by healthcare providers and visitors and the restriction of one visitor per patient. All precautions were observed during data collection. The primary researcher (MZ), who is employed at the hospital, recruited family members during their visits to the ICUs. However, actual data collection occurred once the patient had progressed to the high care or the general units. MZ approached the family members who had indicated an interest in participating in the study whilst in the ICU, orientated them to the study and the information sheet and then proceeded to obtain informed consent. The questionnaires (English or isiZulu, according to their preference) were handed to the family members in a sealed envelope, which was collected by MZ on the following day during visiting hours.

Statistical analysis: The International Business Machines (IBM) Statistical Package for Social Sciences (SPSS) version 27.0 was used for statistical analysis. Descriptive statistics of percentages, means and standard deviations were used to describe the respondents’ demographic data, spiritual practices most and least desired by the respondents and respondents’ total scores of the NSTS. The Pearson chi-square test was used to establish any associations between the respondents total NSTS scores and their sociodemographic variables.

Ethical considerations: The study obtained all relevant approvals and permissions; namely approval from the tertiary institution ethics committee (HSSREC/00000730/2019); ethical approval from hospital ethics committee (approval number: UNI-2019-0045) and gatekeeper approvals for the study and data collection. To obtain informed consent from respondents, full disclosure was provided, they were assured that information provided would be kept confidential and their names were not required in the research documents to ensure anonymity.

## 3. Results

Forty-seven questionnaires were distributed to the respondents and forty-seven completed questionnaires were received, yielding a survey response rate of 100% (47/47). 

### 3.1. Sociodemographic Characteristics of the Respondents

Of the 47 respondents, females dominated the sample at 66.0% (*n* = 31) and over half of the respondents (57.4%, *n* = 27) were Christian. The majority of the respondents (51.1%, *n* = 24) fell in the 31–50-year age group. In relation to the family member’s relationship to the patient, the highest responses (19.1%, *n* = 9) were for siblings and partners (Table 1). 

### 3.2. Preferences of Family Members Regarding Spiritual Care Practices

The extent to which family members desired each spiritual care practice to be provided by an intensive care nurse was determined by measuring the frequencies and percentages of participant’s responses to each statement of a spiritual care nursing practice (Table 2). To reiterate, each spiritual care practice of the NSTS was scored on a four-point Likert scale (1 = strongly disagree to 4 = strongly agree). Results showed that 38.3% (*n =* 18) and 40.4% (*n* = 19) strongly agreed to the items “Help me to have quiet time “and Help me laugh”, while only 10.6% (*n* = 5) of the respondents strongly disagreed to the NSTS items, “Tell me about spiritual resources nearby that I can use” and “Arrange for a chaplain to visit me”.

### 3.3. Most and Least Preferred Spiritual Care Practices of Family Members

Measures of the mean scores of the NSTS items indicated which spiritual care practices were the most and least preferred. The findings revealed that the three spiritual care practice items most preferred by family members were: “*to be helped to have quiet time and space*” (*M* = 3.32, SD = 0.59); “*to*
*help them laugh for example by sharing a joke*” (*M* = 3.30, SD = 0.66); and “*ask me what gives life meaning,”* (*M* = 3.13, SD = 0.68).

The three spiritual care practice items least preferred by family members were for intensive care nurses: “*to pray with them”,* (*M* = 2.77, SD = 0.77); “*to bring them humorous things*,” (*M* = 2.70 SD = 0.72) and lastly, “*to arrange for a chaplain to visit them*,” (*M* = 2.70, SD = 0.91) (Table 3). 

### 3.4. Overall Scores of Family Members Regarding Spiritual Care Practices

Calculation of the overall scores of the NSTS indicated a minimum overall score of 26 and a maximum score of 76, with a mean score of 58. 40 (20–80). Grouping of the overall NSTS scores, indicated that the majority of the respondents, (95.7%; *n* = 45) had a strong desire for the intensive care nurses to provide spiritual practices while 4.3% (*n* = 2) did not desire nurse to provide spiritual practices. 

### 3.5. Associations of the Sociodemographic Characteristics of Respondents with the Total Scores of the NSTS

The cross tabulation performed using the Chi-square test to determine the influence of the sociodemographic characteristics on the respondents’ total scores revealed no significant association, since their results indicated a *p*-value > 0.05, as illustrated in Table 4.

## 4. Discussion

The study provides important information in an area of limited research focus, namely family members’ preferences of spiritual care practices from nurses in ICUs. The sociodemographic findings on gender, age and religious orientation patterned the findings of other studies, where females dominated the sample, (Gabriel, Creedy and Coyne [24]), a greater percentage of the respondents (51.1%) were within the age bracket of 31–50 years of age (Moosavi, Rohani, Borhani and Akbari [25]) and, consistent with the religious composition of the South African population, the majority of the respondents (57.4%) in the current study were Christian [26].

It is worthy to note that the family member-patient relationships varied in the current study, showing the existence of the extended family. This finding may be attributed to the notion of kinship care in South Africa where the extended family also take on the role of family caregivers [27].

We found that the most preferred spiritual care practice was for nurses to provide family members with quiet time and space. The need for undisturbed time during an illness is fundamental for reflection and spiritual rejuvenation, a respite that brings calm to both patients and family members [28]. While other studies have also alluded to the importance of providing a quiet environment for patients and their family members as part of spiritual care, [18,22] the practice may not be possible in an ICU typically filled with sounds and activities [28]. Tedder [29], in awareness of the general chaos of hospital environments, suggests that hospital management could look at biophilic designs of family waiting rooms that may create the spiritual spaces needed by family members. 

The second most preferred spiritual care practice was for nurses to share humor with family members. Our finding was supported by the study of Linge-Dahl, Heintz, Ruch and Radbruch [30] where the item was also among the mostly desired spiritual care therapeutics. The ICU environment is riddled with feelings of distress and despair, and in view of perhaps the optimism and distraction that humor brings to family members, intensive care nurses should look at innovative ways to incorporate humor in their interactions with family members [18]. 

In the current study, the item of wanting nurses to listen to stories about their life events was the third most highly preferred spiritual care practice for family members. The finding is also affirmed in the studies by Karnjuš, Bašić and Babnik [31]; Kanwal, Afzal, Kousar, Waqas and Gilani [32]; Swinton [33], where patients and family members believed that reviewing their life events resulted in a deeper understanding of their illness. Helping patients and family members to find meaning in their illness is believed to assist them with disease acceptance and coping [34]. It should be noted that family members who fail to integrate the meaning and purpose of life during acute illnesses may experience spiritual distress. 

The benefits of integrating art in spiritual care includes that of patients and their family members finding meaning in their illnesses and being able to express their emotions, which are seen to impact on their spiritual well-being [35]. In the current study, and that of Taylor and Mamier, [22] the spiritual care practice of using art was among the three least-desired spiritual care practices (*M* = 2.74 and 2.1, respectively). In contrast, Gardner, Tan and Rumbold [36] found that family members and patients used music as part of spiritual healing. Ettun et al. [37] explained that incorporating arts-based interventions in spiritual care is as important as the conventional care of prayer; however, healthcare providers must ensure that the type of art resonates with the person. 

In their study, Albaqawi et al. [38] mentioned that introducing appropriate humor to the patients and family members was a necessary part of spiritual care. Hardy [39] identified that the use of humor was a positive force in healthcare practice. However, the current study revealed that the idea of the nurses bringing them humorous things was among the three least-desired spiritual care practices by family members. Intensive care nurses should be circumspect concerning the appropriateness of humor for both the family and the circumstances. 

Interestingly, the need for the nurses to arrange for a chaplain to visit family members was a least preferred practice in the current study. This is contrary to the findings of studies by Hennessy et al. [40] where participants reported that chaplains provided support during the illness of family members, yet similar to the findings of Teague et al. [41] where family members expressed that hospital chaplains appeared disinterested in their plight. We believe that family members must be encouraged to choose the person/s to provide spiritual support, as it is probably the spiritual care skills of the person that informs their choice rather than the profession of the person. 

The overall mean NSTS score indicated that there was a strong preference among family members for nurses to provide them with spiritual care in the ICUs. The finding was similar to that of the NSTS score in Taylor and Mamier’s [22] study, but significantly higher in the current study, having a mean of 58.40 versus a mean of 47.13. 

At the same time, some family members may not want to engage with nurses in spiritual care activities and nurses must offer support according to the desires of family members. 

The current study findings of there being no associations between gender, age and relationship to the patient and total NSTS scores were again similar to those of Taylor and Mamier [22]. The latter study concluded a weak positive correlation between how often respondents attended religious services and their total NSTS scores. The sociodemographic variable was not included in the current study. 

### 4.1. Implications for Clinical Practice 

Given that most spiritual interventions offered to family members are designed from the perceptions of healthcare providers, it is recommended that comparative studies of family members’ preferences for spiritual care practices and healthcare providers’ perceptions of families’ needs for spiritual care practices be conducted. The findings could possibly lead to evidence-based, focused and appropriate spiritual care interventions being designed for family members. The current study revealed that spiritual care practices that are the norm of spiritual care provided to patients and family members do not correlate well with the actual preferences of family members. Hence, nurses should engage with family members through spiritual interviews, or discussions to elicit individual preferences on spiritual care. 

### 4.2. Study Limitations 

The onset of the COVID-19 pandemic resulted in a small sample size, in one hospital, therefore limiting representation to the general family member population. However, the methodology was applied rigorously and the researchers wanted to share the findings of essential research conducted in the field of spirituality in intensive care units. We therefore recommend that similar studies with larger samples be conducted with both family members and patients in the ICUs, within the same hospital and in other hospitals.

## 5. Conclusions

The study revealed that family members with close family members in ICUs strongly desired that nurses provide them with various spiritual care practices. Specifically, family members preferred nurses to provide spiritual spaces, humor and assist them review life events. Spiritual care practices that the family members least preferred related to nurses assisting them to illustrate their spirituality, bring items that were considered humorous and arrange for the visits of religious leaders. We conclude that spiritual care practices are an important part of holistic nursing care in ICUs and that the family members’ preferences should be the compass that directs their spiritual care.

## Figures and Tables

**Table 1 healthcare-10-00595-t001:** Sociodemographic variables of respondents (*n* = 47).

Characteristics	Character Specification	Freq (*n*)	Percent (%)
Gender	Male	16	34.0
	Female	31	66.0
Age	18–30	15	31.9
	31–50	24	51.1
	>50	8	17.0
Religion	Christian	27	57.4
	Other	8	17.0
	Hindu	7	14.9
	Muslim	3	6.4
	Buddhist	2	4.3
Relationship to the patient	Partner	9	19.1
	Sibling	9	19.1
	Spouse	8	17.0
	Other (Aunt, Uncle, Grandparent, Cousin)	8	17.0
	Parent	6	12.8
	Child	5	10.6
	Guardian	2	4.3

**Table 2 healthcare-10-00595-t002:** Description of family members’ responses to the NSTS (*n* = 47). (Frequencies (Freq), %).

Item: In General, I Would Want My Family Member’s Nurse to:	Strongly Disagree (1)		Disagree (2)		Agree (3)		Strongly Agree (4)	
	Freq	%	Freq	%	Freq	%	Freq	%
Help me to have quiet time and space	0	0	3	6.4	26	55.3	18	38.3
Help me laugh (e.g., share a joke)	0	0	5	10.6	23	48.9	19	40.4
Ask me about what gives life meaning	1	2.1	5	10.6	28	59.6	13	27.7
Offer to talk to me about meditation	1	2.1	8	17.0	23	48.9	15	31.9
Ask me about how I relate to God (or the Ultimate Other)	1	2.1	11	23.4	21	44.7	14	29.8
Listen to me talk about my spiritual strengths	2	4.3	9	19.1	22	46.8	14	29.8
Listen to stories of my life	2	4.3	9	19.1	25	53.2	11	23.4
Help me, if needed, with my religious practices	2	4.3	11	23.4	21	44.7	13	27.7
Listen to me talk about my spiritual concerns	4	8.5	7	14.9	25	53.2	11	23.4
Offer to talk to me about the difficulties of praying when sick	3	6.4	8	17.0	28	59.6	8	17.0
Ask me about my spiritual beliefs	4	8.5	8	17.0	25	53.2	10	21.3
Arrange for my minister or a spiritual mentor to visit me	2	4.3	15	31.9	19	40.4	11	23.4
Ask me about religious practices	3	6.4	11	23.4	25	53.2	8	17.0
Tell me about spiritual resources nearby that I can use	5	10.6	7	14.9	27	57.4	8	17.0
Offer to pray privately for me (i.e., nurse prays for me later while alone)	1	2.1	16	34.0	23	48.9	7	14.9
Help me to think about my dreams	3	6.4	12	25.5	25	53.2	7	14.9
Offer to pray with me	4	8.5	12	25.5	22	46.8	9	19.1
Teach about ways to draw or write about my spirituality	2	4.3	15	31.9	23	48.9	7	14.9
Bring me humorous things	2	4.3	15	31.9	25	53.2	5	10.6
Arrange for a chaplain to visit me	5	10.6	13	27.7	20	42.6	9	19.1

**Table 3 healthcare-10-00595-t003:** Mean NSTS scores on preferred spiritual practices (*n* = 47).

Item: In General, I Would Want My Family Member’s Nurse to:	Mean	±SD
Help me to have quiet time and space	3.32	0.59
Help me laugh (e.g., share a joke)	3.30	0.66
Ask me about what gives life meaning	3.13	0.68
Offer to talk to me about meditation	3.11	0.76
Ask me about how I relate to God (or the Ultimate Other)	3.02	0.79
Listen to me talk about my spiritual strengths	3.02	0.82
Listen to stories of my life	2.96	0.78
Help me, if needed, with my religious practices	2.96	0.83
Listen to me talk about my spiritual concerns	2.91	0.86
Offer to talk to me about the difficulties of praying when sick	2.87	0.77
Ask me about my spiritual beliefs	2.87	0.85
Arrange for my minister or a spiritual mentor to visit me	2.83	0.84
Ask me about religious practices	2.81	0.80
Tell me about spiritual resources nearby that I can use	2.81	0.85
Offer to pray privately for me (i.e., nurse prays for me later while alone)	2.77	0.73
Help me to think about my dreams	2.77	0.79
Offer to pray with me	2.77	0.87
Teach about ways to draw or write about my spirituality	2.74	0.77
Bring me humorous things	2.70	0.72
Arrange for a chaplain to visit me	2.70	0.91

**Table 4 healthcare-10-00595-t004:** Associations between sociodemographic variables and total scores in the NSTS.

Sociodemographic Variable		Total Score for NSTS	
	Pearson Chi-Square Value	df	Sig (*p* Value)
Gender	1.078	1	0.54
Age	0.570	2	1.00
Relationship to patient	5.070	6	0.50
Religion	2.327	4	0.48

## Data Availability

Not applicable.

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
