# Peer review of "Spiritual Care: A Description of Family Members’ Preferences of Spiritual Care Nursing Practices in Intensive Care Units in a Private Hospital in Kwa-Zulu Natal, South Africa"

_healthcare, 2022, doi:10.3390/healthcare10040595_

Round 1
Reviewer 1 Report
The abstract should be rewritten. The results of the study should not be in the abstract. When the results are in the abstract, there is no need to read the entire article.
The importance of recognizing not all patients ascribe to the same spiritual practices and the need for nurses to address the spiritual needs of their patients as unique and individually as their patient.
The topic is relevant as nurses are required to address the spiritual needs of their patients but many nurses struggle to know exactly how to do this. The article adds to the body of knowledge that addresses the importance of spirituality in healthcare.
methodology: With a small n= for a qualitative study perhaps a mixed methodology using qualitative methods may have yielded a more robust study and analysis.
Author Response
Thankyou and please see attachment

Reviewer 2 Report
- The article presents an important issue concerning the spiritual care and support in the intensive medical care. The title is well-formulated and it accurately communicates the content of the text.
-
I suggest a small modification to the summary. The frequencies and statistics indicated concern the whole group, so it is justified to use capital letters to mark size (number of participants, N) and mean values (M). It is also worth adding standard deviations (SD) to the means. When it comes to the substantive value of the summary, it is correct and includes the necessary information.
- The studied group is relatively small (N=47), so it is worth considering this research as a pilot study or making a clear reference to the small size of the study group. Perhaps this is due to the pandemic, which was mentioned in Limitations.
-
Introduction outlines well the subject matter of the article. I suggest changing the term ‘loved one’ to ‘close one’ in line 30.
-
Research procedure and the selection method of the study group were correctly characterised.
-
The description of the applied research tool Nurse Spiritual Therapeutic Scale [NSTS] should be supplemented by the presentation of sample items of this scale.
- Sociodemographic characteristics of the respondents should be completed with mean age of the study participants and its standard deviation.
-
Research results were clearly presented and described. Discussion is very interesting, thorough and well-written. Conclusions are prepared well.
Author Response
Thankyou and please see attachment
